# Superelastic NiTi Functional Components by High-Precision Laser Powder Bed Fusion Process: The Critical Roles of Energy Density and Minimal Feature Size

**DOI:** 10.3390/mi14071436

**Published:** 2023-07-18

**Authors:** Shuo Qu, Liqiang Wang, Junhao Ding, Jin Fu, Shiming Gao, Qingping Ma, Hui Liu, Mingwang Fu, Yang Lu, Xu Song

**Affiliations:** 1Department of Mechanical and Automation Engineering, Chinese University of Hong Kong, Shatin, Hong Kong, China; qushuo@link.cuhk.edu.hk (S.Q.); jhding@link.cuhk.edu.hk (J.D.); shiminggaozju@gmail.com (S.G.); qpma@cuhk.edu.hk (Q.M.); 1155168493@link.cuhk.edu.hk (H.L.); 2Department of Mechanical Engineering, City University of Hong Kong, Kowloon, Hong Kong, China; liqiawang2-c@my.cityu.edu.hk; 3Department of Mechanical Engineering, The Hong Kong Polytechnic University, Hung Hom, Kowloon, Hong Kong, China; jin0103.fu@connect.polyu.hk (J.F.); ming.wang.fu@polyu.edu.hk (M.F.); 4Department of Mechanical Engineering, The University of Hong Kong, Pokfulam, Hong Kong, China

**Keywords:** 3D printing, laser powder bed fusion, NiTi alloy, energy density, TPMS lattice, robotic cannula, mechanical testing

## Abstract

Additive manufacturing (AM) was recently developed for building intricate devices in many fields. Especially for laser powder bed fusion (LPBF), its high-precision manufacturing capability and adjustable process parameters are involved in tailoring the performance of functional components. NiTi is well-known as smart material utilized widely in biomedical fields thanks to its unique superelastic and shape-memory performance. However, the properties of NiTi are extremely sensitive to material microstructure, which is mainly determined by process parameters in LPBF. In this work, we choose a unique NiTi intricate component: a robotic cannula tip, in which material superelasticity is a crucial requirement as the optimal object. First, the process window was confirmed by printing thin walls and bulk structures. Then, for optimizing parameters precisely, a Gyroid-type sheet triply periodic minimal-surface (G-TPMS) structure was proposed as the standard test sample. Finally, we verified that when the wall thickness of the G-TPMS structure is smaller than 130 μm, the optimal energy density changes from 167 J/m^3^ to 140 J/m^3^ owing to the lower cooling rate of thinner walls. To sum up, this work puts forward a novel process optimization methodology and provides the processing guidelines for intricate NiTi components by LPBF.

## 1. Introduction

NiTi, namely Nitinol, is a unique functional alloy known for its superelasticity and shape-memory property, which result from the reversible phase transformation between austenitic B2 and martensitic B19’ phases [1]. It is widely used in aerospace [2], medical biology [3,4,5], and other fields [6,7,8]. However, owing to the low thermal conductivity and poor machinability [9], traditional fabrication methods can only manufacture NiTi into simple geometric shapes [10], which cannot satisfy the requirements of intricate components with complex geometries.

Additive manufacturing (AM) has recently been employed for NiTi alloy fabrication [11,12,13,14]. Among all the AM methods, laser powder bed fusion (LPBF) is ideal for achieving intricate components with complex geometry [15,16] and high fidelity [17]. The process parameters of LPBF can greatly affect the properties of the printed NiTi components [15,18]. The scan speed, laser power, and hatch space are often adjusted to control the composition and microstructures of printed components [15,16,19]. Some researchers have also manipulated the energy density to design the phases and performance of NiTi alloys [20,21]. However, these studies focus on the process parameter optimization for bulk materials, and the study on the influence of energy density on intricate components and thin wall structures is lacking. The difficulty is that the different wall thicknesses and strut diameters of those components result in various thermal histories and cooling rates [22], which lead to different material microstructures from the bulk material by the same process parameters. Therefore, conventional test specimens, such as cubes, cylinders and dog bones, are unable to represent the property of NiTi intricate components with thin walls and complex structures accurately. The process parameters, geometrical shapes, and minimal feature sizes should be considered holistically in a comprehensive manner.

Moreover, compared to conventional LPBF, the high-precision laser powder bed fusion (HP-LPBF) method has been recently proposed to fabricate intricate components [17,23,24]. It combines fine beam size (25 μm), fine powders (0–25 μm), and thin layer thickness (10 μm) together and results in different microstructures (e.g., grain sizes, dislocation densities, and molten pool sizes), the faster cooling rate of the molten pool, lower surface roughness, higher resolution, and smaller distortion [25]. Researchers have also used HP-LPBF to achieve high-performance NiTi components. Fu et al. [23] employed HP-LPBF to fabricate NiTi samples to study their thermal and mechanical behaviors. However, the microstructural inhomogeneity of as-printed samples resulted in a suppressed martensitic transformation owing to the unexpected precipitation. It could be found that the NiTi fabrication is extremely sensitive to the manufacturing process, including raw material [26], process parameters [27,28], as well as minimal feature size [22]. Xiong et al. [29] used HP-LPBF to successfully achieve the strut lattices with excellent shape recovery. However, the optimal process parameters for lattices with inclined structures were acquired by printing thin wall and grate structures vertically. It is somehow unreasonable because the thermal diffusion of the molten pool varies with the inclination angles [30], especially for NiTi alloy.

In general, intricate components often suffer from complicated stress conditions [31], such as local compression, tension and bending. For example, when the helical robotic cannula tips bend [31], the outer parts are in tension, and the inner parts are in compression. In addition, the snake-like NiTi actuator [32] and NiTi spring [33] are compressed and stretched simultaneously at various locations in the thermal and mechanical responses. The struts of NiTi stents are also under compression and tension when supporting the bending deformation [34]. Moreover, NiTi alloys have asymmetric superelastic behavior in tension and compression modes of deformation [35], which is ascribed to the quicker martensite evolution rate under tension [36]. Thus, a comprehensive mechanical property evaluation of NiTi is necessary for intricate component fabrication. However, conducting mechanical testing (such as compression or bending) on thin wall structures is hard since it will result in the buckling of structures. Furthermore, it is unrealistic to directly fabricate and test intricate components for parameter optimization due to the high cost and low efficiency.

Therefore, in order to obtain the optimal process parameters for NiTi intricate components, a triply periodic minimal surface (TPMS) sheet structure was proposed in this work as the standard test geometry. It is a class of periodic open-cell shell lattices with large specific areas and smooth surfaces. These sheet structures were created by offsetting the zero-level mid-surface. Thus, the wall thickness could be varied easily by adjusting the level parameters of the implicit mathematical expressions. In contrast to skeleton-based TPMS, sheet-based TPMS shows superior mechanical properties for avoiding local failure and owns smaller minimal feature sizes (such as wall thickness) due to the higher surface-to-volume ratio [37]. In this case, the local failure of superelastic NiTi intricate components means not only material fracture but also the generation of irreversible martensite transition and loss of pseudo-elasticity. Among all sheet TPMS structures, Gyroid-type sheet TPMS (G-TPMS) is the most isotropic lattice structure in mechanical response [38]. Besides, G-TPMS has the least stress fluctuations during compression, which means it possesses a more uniform stress distribution and less chance for local failure among all TPMS lattice structures. Therefore, it can be employed as the standard test sample to predict the material performance in the NiTi intricate components.

In this study, the effects of the process parameters on the printing dimensions, microstructure, roughness, phase transformation behavior, and mechanical properties of NiTi by HP-LPBF were studied. Cylinders and G-TPMS structures with different wall thicknesses (65, 130, and 260 μm) were printed with different energy densities. It is demonstrated that when the components possess small wall thickness, the energy density should be reduced accordingly to balance the effect of decreasing the cooling rate to achieve good superelasticity. Finally, robotic cannula tips were fabricated and achieved full recovery with bends greater than 90 degrees at room temperature, which built the knowledge gained from the G-type TPMS sample fabrication.

## 2. Materials and Methods

### 2.1. Materials and Processes

The AM fabrication was performed using a HP-LPBF machine (Han’s Laser, M100μ, Shenzhen, China). The protective atmosphere was argon with an oxygen content of less than 500 ppm. The laser source had a wavelength of 1070 nm. The substrate material was NiTi in nearly equal proportions. The beam size was 25 μm, and the layer thickness was 10 μm. NiTi powder was provided by the GKN Hoeganaes Corporation, Cinnaminson, NJ, USA. The particle size of the as-received powder ranged from 5 μm to 25 μm, and the size distribution of the powder is shown in Table 1. D_10_ indicates the diameter at 10% cumulative volume. D_50_ means the median diameter, and D_90_ means the diameter at 90% cumulative volume. The chemical composition of the powder is listed in Table 2. Thin-walled samples were printed using a single laser track with a length of 5 mm. Cube samples with dimensions of 5 mm × 5 mm × 5 mm were used to characterize the relative density (RD) and surface roughness. The size of the tensile samples was described in detail in the previous study [25]. The compression samples had a diameter of 2.6 mm and a height of 3 mm. The hatch distance was 50 μm, and the angle of rotation between each layer was 67° to reduce thermal residual stress [23]. Each layer was filled with laser tracks in a zigzag pattern. Thin-walled samples were fabricated using different powers and scan speeds, as shown in Table 3. The linear energy density (LED) was calculated as P/V, where P is the laser power, and V is the scan speed. The bulk samples fabricated using different parameters were named as shown in Table 4. The volumetric energy density (VED) was calculated as P/Vht, where h is the hatch distance of 50 μm, and t is the layer thickness of 10 μm. The scanning path was generated by Materialise Magics^®^, Leuven, Belgium. Figure 1 shows the models and scanning path of thin wall bulk with a hatch distance of 50 μm, which presents the LED/VED distribution with process parameters. After fabrication, the as-printed samples were cut from the substrate using wire electrical discharge machining (EDM).

### 2.2. Characterization

Scanning electron microscopy (SEM) observations were conducted using a JCM-6000Plus instrument (JEOL Ltd., Tokyo, Japan). Optical microscopy (OM) observations and roughness measurements were conducted using an RH-2000 high-resolution 3D microscope (Hirox–USA, Inc., Hackensack, NJ, USA). The RD values of the samples were measured using the Archimedes method. Three NiTi samples per parameter set were weighed both in the air and in an ethanol medium using an analytical balance with an accuracy of 0.01 mg. To observe the morphology of the molten pools, the side surfaces of the samples were ground, polished, and etched using a solution containing 5% HF, 15% HNO_3_, and 80% H_2_O by volume fraction. X-ray diffraction (XRD) patterns, obtained using a high-resolution Rigaku SmartLab X-ray diffractometer, were used to determine the phase compositions of the different samples. The 2θ angle range was from 20° to 100°. A Mettler Toledo differential scanning calorimetry (DSC) was used to measure the phase transformation temperatures (TTs) of the as-printed NiTi samples and powder at heating and cooling rates of 10 °C/min. Tensile tests were conducted under quasi-static speed at a strain rate of 10^−3^ s^−1^. The compression test was conducted at a strain rate of 10^−3^ s^−1^ during strain control loading and at a rate of 100 N/s during stress control loading. Mechanical tests were conducted on the three samples fabricated with each parameter set.

### 2.3. Geometric Modeling of Component Structures

A robotic cannula designed by Yan et al. [31] for a minimally invasive intracerebral hemorrhage (ICH) evacuation is the focus of the current study, which includes an actuation platform, pre-curved cannula body and flexible cannula tip. As one of the design criteria, the cannula tip, as shown in Figure 2a, requires at least a 90° bending capability in any direction for aspirating the peripheral ICH for clinical decompression [39], for which good superelasticity of NiTi material is required.

The models with the TPMS structure, generated using MATLAB software R2020a, were subjected to compression testing. The G-TPMS cube was filled with four unit cells of a G-TPMS, which can be considered as an infinite array in the mechanical behaviors [40,41], as illustrated in Figure 2c. Three groups of G-TPMS cubes with different relative densities (10%, 20% and 40%) are presented in Table 5. The shell thickness was set as 65, 130, and 260 μm, respectively, and the unit cell is 2 mm. The G-TPMS lattice was defined by the following function [42]:(1)$$\begin{array}{c} \phi_{G}\left(x,y,z\right)=\sin\left(wx\right)\cos\left(wy\right)+\sin\left(wz\right)\cos\left(wx\right)+\sin\left(wy\right)\cos\left(wz\right)=C \end{array}$$
where $w=\frac{2\pi }{L}$, L indicates the periodic length of TPMS, and C controls the TPMS interface. Different thicknesses were obtained by varying C. The geometric model was rescaled for the fabrication of LPBF into the target unit cell size.

To reveal local stresses, simulations on the cannula tip and TPMS shell lattices were performed using finite element (FE) modeling. The mechanical response of G-type TPMS under compression was analyzed using the finite element package Abaqus/Explicit 2017-1. Figure 2b,d show the FE model setup of TPMS and tip. TPMS structure is implemented by modeling and connecting rigid plates to its top and bottom nodes to provide the same condition of experimental tests. All degrees of freedom of the bottom plate are fixed, and the strain rate of the top plate is 10^−3^ s ^−1^. An average cell size of 0.2 mm forms the mesh for TPMS shells made from linear triangular prism elements (C3D6) [38]. The cannula tip is meshed by an average cell of 0.1 mm with four-node tetrahedral elements (C3D4). All degrees of freedom of the bottom of the tip are fixed. All points of the tip’s top surface are coupled on the center point of the bottom surface and rotated at an angle of 90 degrees. Semi-automatic mass scaling is configured as a minimum target time increment of 5 × 10^−4^ s for providing a low ratio of kinetic energy to internal energy (<10%) during compression. The tensile property of NiTi produced using P7V6 for FE is shown in Appendix A. The other details can be referred from the previous report [38].

## 3. Results and Discussion

### 3.1. Effect of Process Parameters on Thin-Wall Structure Fabrication

The morphology of the powder was observed using SEM (Figure 3a,b). The phase composition of the powder was determined by XRD, as shown in Figure 3c. The phase-TTs were measured using DSC, as shown in Figure 3d. The R-phase was detected before the martensite phase during the heating and cooling process. The R-phase transformation peak is caused by the thermoelastic transition between the austenite and R-phases. The formation of the R-phase has been attributed to crystallographic defects. Thin-walled structures were first printed to evaluate the printing resolution of HP-LPBF and the influence of parameters on molten pool size and microstructure. Figure 4 shows the characterization of thin-walled structure (TWS) samples fabricated using different parameters in Table 3. Figure 4a illustrates the top cross-sectional morphologies of samples. All TWS samples were printed layer-by-layer using a single laser track. Therefore, the wall thickness was the smallest printing size for the corresponding process parameter. The wall thickness was measured, as shown in Figure 4a. The TWS wall thickness results are shown in Figure 4b. The highest resolution of 62 μm was achieved in L50, using a laser power of 50 W and a scan speed of 1000 mm/s.

Moreover, the wall thickness increased with increasing LED (Table 3). Figure 4c shows the XRD diffraction patterns of the groups of low- (L83–L117) and high-speed samples (L50–L70). Similar diffraction patterns are observed for these samples, which indicates that parameter variation has a limited effect on phase composition. The B19′ NiTi phase was the dominant phase. The Ni_3_Ti phase, which consumes the Ni content of the matrix, could also be discerned. The secondary Ni-rich phase resulted from precipitation during the layer-by-layer fabrication process. In addition, peaks of a Ti-rich phase (Ti4Ni_2_Ox/Ti_2_Ni) and Ni_4_Ti_3_ were also detected, which resulted from microstructural inhomogeneity [43]. Figure 4d illustrates the thermally induced phase-transformation behavior of L50–L117, fabricated with different LEDs, respectively, as analyzed using DSC. The austenitic and martensitic transformation peak temperatures (Ap, Mp) of samples were increased by the increase of LED due to the evaporation of Ni element. The TTs of the TWS samples are similar to those of the original powder shown in Figure 3. Therefore, it can be concluded that the loss of Ni content is low during TWSs fabrication. Generally, in TWS fabrication, it is demonstrated that 62 μm of resolution and controllable microstructure of TWS can be realized by HP-LPBF.

### 3.2. Effect of Process Parameters on Bulk Fabrication

To verify the differences between bulk and thin wall, the cubes were printed to demonstrate the variety of microstructures and phases by VED. Cubes were fabricated using different process parameters given in Table 4. The bulk densification optimization is shown in Appendix A. When the VED is less than 160 J/mm^3^, the NiTi alloy cube can be printed with RD > 99.5%, which is considered highly dense [23].

Figure 5 presents the microstructural characterizations of the cube samples. Figure 5a,b show the depths and half-widths of molten pools in the cubes printed with laser powers of 50–70 W and scan speeds of 600–1000 mm/s. Obviously, molten pool depth and width decreased with increasing scan speed and decreasing laser power. The measurement method is shown in the insets. In Figure 5d,i, the polished and etched side surfaces of the cubes show the molten pool morphologies. Vertical cracks occurred in the middles of the molten pools; pores appeared at the bottoms. These pores could be attributed to the keyhole fluctuation in the fabrication process [44,45]. Figure 5c shows the molten pool depth-to-width ratio (D/W) obtained by different printing parameters. Based on the side-surface morphologies, the samples containing pores and cracks were marked with red circles, while dense samples were marked with blue circles. It can be concluded that when D/W exceeded 1.8, the melting condition became unstable [44], and pores and cracks appeared. Appendix A shows the relationship between D/W and RD. It can be concluded that the RD decreased sharply when D/W was larger than 1.8. In terms of surface roughness, Appendix A displays the results of the top and side surface roughness of the as-printed cubes. Top surface Ra values were from 0.4 μm to 1.45 μm, as presented in Appendix A. Side surface Ra values for side roughness were between 6.48 and 10 μm, as presented in Appendix A. It demonstrated that the process parameter used here can build dense bulk samples with stable molten pools, and this is because no spatter was observed on the top surface. In regard to spatters on the side surface, they were formed due to the interaction between powders and high-temperature metal liquid and thus had an average size of about half of the powder size [46].

Figure 6a illustrates the XRD patterns of the as-printed NiTi samples obtained using different parameters. The B2 and B19′ phases were detected in all samples. The main secondary phases changed with increases in the VED. Ni-rich phases of Ni_4_Ti_3_ and Ni_3_Ti were observed in cubes fabricated with low energy densities. This is attributed to the lower temperature of the molten pool and faster cooling rate enabled by energy densities of <150 J/m^3^, which lead to less Ni loss [47]. When the VED was greater than 150 J/m^3^, the Ti_4_Ni_2_Ox/Ti_2_Ni phase appeared. When the VED was greater than 200 J/m^3^, peaks of the Ni-rich phases of Ni_4_Ti_3_ and Ni_3_Ti could not be detected because of the large loss of Ni in the high-energy-density LPBF process. Figure 6b illustrates the phase transformation behaviors of the as-printed NiTi samples fabricated using different parameters. The phase TTs of all cubes (Ap = 85–95 °C; Mp = 52–60 °C) were higher than those of the TWS samples and original powder owing to their greater loss of Ni, which resulted from the overlapped molten pools in the printed bulk, as shown in Figure 5. Moreover, the austenitic and martensitic peak temperatures increased with increasing VED, as shown in Figure 6c. The relationship between temperature and VED was determined by linear fitting. The corresponding equations of linear fit between Ap or Mp and VED are given in Figure 6c.

Overall, phase compositions are varied with energy densities, which do not occur in TWS. It shows that the overlapping of molten pools in bulk printing and different thermal field makes the relationship of bulk between parameters and properties different from that of TWS. Hence, when the minimal feature size decreases down to several molten pool sizes, as shown in Figure 2, the relationship between parameters and properties is different for TWS and bulk structure. In addition, it is hard to conduct mechanical experiments on TWS, such as compression tests. Buckling is prone to occur in the bending/compression testing. Moreover, the tolerance of defects (cracks, pores, and unmelted powders) is also low for TWS mechanical testing. Hence, it is necessary to use a new structure to evaluate the influence of parameters on the NiTi’s intricate components.

### 3.3. Superelastic Property of NiTi

#### 3.3.1. Effect of Process Parameters on Bulk Samples

Figure 7 depicts the loading-unloading compression results of the P5V6, P5V10, P7V6, and P7V10 samples. The compression test was conducted by two types of modes (same stress and same strain). Figure 7a,c shows the representative stress-strain curves of four parameters. Figure 7b,d show irreversible/recovered strain and recovery ratio of different samples arranged from low to high VED. Irreversible strain means the residual strain after the unloading process. Recovered strain indicates the resilient strain during the unloading process. The recovery ratio is defined as the ratio of recovered strain to total strain (recovery strain + irreversible strain). As a result, P5V6 (167 J/m^3^) and P7V10 (140 J/m^3^) samples exhibited better elastic recovery properties than P5V10 (100 J/m^3^) and P7V6 (233 J/m^3^) samples in two types of modes (same stress and same strain). P5V6 and P7V10 samples showed a recovery ratio higher than 0.35, and P5V10 and P7V6 samples depicted a recovery ratio lower than 0.30 under the same compression stress. In terms of the same compression strain experiment, P5V6 and P7V10 samples showed a recovery ratio higher than 0.40, and P5V10 and P7V6 samples showed a recovery ratio lower than 0.35. Overall, P5V6 samples possess the best shape recovery property. From Figure 6, it could be found that there are Ni_4_Ti_3_ and Ti_2_Ni phases in P5V6 cube samples. It is well known that the presence of Ni_4_Ti_3_ is beneficial to the superelastic property of NiTi components [48]. A lack of energy input leads to the heating temperature and duration not being sufficient to allow the Ni_4_Ti_3_ precipitation to fully form. Thus, the Ni_4_Ti_3_ precipitation declines in the samples by P5V10 [49]. The P7V10 and P5V6 samples were printed by higher energy densities with higher content of Ni_4_Ti_3_ precipitation. However, regarding the P7V6 samples, the Ti_2_Ni phase precipitated excessively, and the Ni element evaporated, which is harmful to superelasticity. Generally, good control of VED for the balance of Ni_4_Ti_3_ segregation and Ni evaporation was the key to the deformation recovery property.

#### 3.3.2. Effect of Process Parameters on TPMS Samples

FE simulation was conducted first to estimate the stress distribution of G-TPMS and cannula tip. Firstly, the tensile properties of NiTi fabricated with different parameters are shown in Appendix A, which were used for FE simulation. The phase transformation and work hardening stages are similar for the variety of power shown in Appendix A. Moreover, Appendix A presents that higher scan speed leads to a lower work hardening rate owing to fewer precipitations in lower VED. P7V6, which exhibits the highest strength and elongation, is used for simulation. Figure 8a–c shows the stress distribution of G-type TPMS lattice under 6% compressed strain. The tensile stress (red and yellow) and compressive stress (blue and green) are distributed in a regular periodic pattern on the surfaces and internal structures of the lattice (Figure 8a,b). Typically, the regions with larger inclination angles possess larger maximum principal stress and therefore play a more significant role in bearing external loads than those with smaller inclination angles. Moreover, the lattice tends to exhibit contrary tensile and compressive stress states on the two sides of the surface, which indicates an obvious bending behavior (Figure 8c). Furthermore, Figure 8d–f present the stress distribution of the cannula tip with a bend. A similar stress distribution on the surface was also observed in the bending tip. Appendix A shows the tension and compression stress conditions of all elements extracted from the FE simulation of TPMS and cannula tip. As a result, the overall stress distribution for TPMS and tip is composed of similar tension and compression stress status. Thus, the TPMS structures under compression can well represent the stress condition of cannula tips with a bend.

Figure 9 shows the as-printed TPMS lattice structures of NiTi with RD of 10%/20%/40%. To evaluate the effects of parameters on samples with different RD, all the lattice structures were fabricated using P5V6/P5V10/P7V10. Figure 9d–f illustrates the top surface of the three structures. The wall thickness also varied with cell size, at 65, 130, and 260 μm characterized by SEM. Additionally, the G-TPMS structures failed to be printed using P7V6 due to its excessive energy input during fabrication, which led to the distortion of thin-walled structures. It verified that the optimal process parameters of bulk printing cannot be directly used for intricate component printing.

Figure 10 depicts the shape recovery properties of TPMS with different RDs using stress or strain control. It can be found that the recovery property of P5V10 is the worst one in the same stress or strain compression experiments due to less precipitation of Ni_4_Ti_3_ [49]. The deformation recovery properties of P7V10 and P5V6 are approximately equal in performance, and P5V6 samples exhibit a better recovery ratio in line with the cylinder compression results shown in Figure 7. However, P7V10 is better for TPMS with RD of 10%. This is because the heat-affected zone and thermal gradient, which are affected by the thermal diffusion of molten pools, will severely influence the precipitation of the second phase (Ni_4_Ti_3_, Ti_2_Ni).

It is well known that the thermal diffusion coefficient of powder is much lower than as-printed solid parts [24]. Therefore, the cooling rate will be lower when the wall thickness is smaller. For TPMS sheet structures, this equals the condition of a lower RD because thin walls have more surfaces in contact with the powder and fewer surfaces in contact with the solid. Hence, the P5V6 parameter will result in more Ni content evaporation and Ti_2_Ni precipitation in 10% RD than 40% RD. Furthermore, P7V10 (140 J/m^3^) owns less VED than P5V6 (167 J/m^3^), which will generate a similar thermal field and precipitation condition for thin wall printing to that of P5V6 for bulk printing. Additionally, P5V6 and P7V10 result in identical recovery properties for TPMS with RD of 20% (130 μm). Overall, when the structure wall thickness is less than 130 μm, P7V10 is clearly better than P5V6 for its shape recovery property.

### 3.4. Superelastic Performance of Robot Cannula Tips: A Case Study

Robotic cannulas are used for minimally invasive intracerebral hemorrhage (ICH) evacuation [31]; such structures require high resolution, low roughness, and excellent superelasticity to achieve at least 90-degree bending capability in all directions for dexterous tip motion [39]. From the results of the process parameter optimization, it can be concluded that the VED influences the shape recovery properties of the as-printed parts. The cannula tip is the structure spiral type with a 120 μm strut diameter designed for ICH. Thus, P7V10 is more suitable to print this component to obtain better superelasticity. The morphology of the cannula tip was characterized using OM and SEM, as shown in Figure 11. It can be seen that the holes for the actuation cable manipulation and helix structure of the tip were successfully fabricated at high resolution. Figure 12a–d illustrate that the cannula tip printed with P7V10 parameters exhibited excellent superelastic properties (>90° of bending) at room temperature (see Appendix A). Although the Mp of the part printed using P7V10 parameters was higher than room temperature, as shown in Figure 6, the matrix also exhibited an austenitic phase in the XRD patterns, and the Ms was close to room temperature. Figure 12e–h show that the cannula tip fabricated with P5V6 parameters remains an unrecovered deformation with a degree of 10.2° after bending (see Appendix A). It is attributed that the optimal parameter for the minimal feature size (120 μm) of a tip is P7V10, shown in Figure 10. Hence, cannula tips with optimal parameters manifest excellent superelastic performance. HP-LPBF is, therefore, a promising method for fabricating high-performance and high-precision NiTi intricate components.

## 4. Conclusions

Superelastic NiTi intricate components were successfully fabricated by HP-LPBF in this study. Extensive parameter studies were conducted to reveal their influences on the microstructure and mechanical properties of as-printed components with different minimal feature sizes (thin wall structure, bulk samples, TPMS structures, etc.). TPMS structures were proposed and verified to be a good candidate as the standard test for parameter study of printing NiTi intricate components and investigation of NiTi thin wall structure mechanical property. Finally, a robotic cannula tip with fine minimal feature size and high superelasticity was fabricated using HP-LPBF as a case study. The main conclusions of this study are as follows:(1)Thin-walled structures were fabricated using parameters that yielded different LEDs. The LED had a significant influence on the wall thickness. The thinnest single track of 62 μm was achieved with a laser power of 50 W and a scan speed of 1000 mm/s.(2)Bulk components with an RD higher than 99.5% could be printed with a VED lower than 160 J/mm^3^. Pores and cracks were observed in the microstructure when the D/W ratio of the molten pools was larger than 1.8. The precipitated phase changed from Ni_4_Ti_3_ and Ni_3_Ti to Ti_4_Ni_2_O_x_/Ti_2_Ni with the increase in VED, which also caused an increase in the TTs.(3)Cylinders by VED of 140~167 J/m^3^ had better shape recovery properties than other parameters (100 J/m^3^ or 233 J/m^3^) due to a good controlling of VED for the balance of Ni_4_Ti_3_ segregation and Ni evaporation.(4)When the wall thickness of the G-TPMS lattice is smaller than 130 μm, the optimal VED for superelasticity is changed from 167 J/m^3^ to 140 J/m^3^ owing to the lower cooling rate of thinner walls.(5)The robotic cannula tips with a 120 μm strut diameter fabricated with 140 J/m^3^ VED exhibited excellent superelasticity properties thanks to the tailored phase compositions. It reveals the coupling effects of parameters, microstructure and feature size in LPBF, and proposes a new approach to processing high-performance intricate NiTi components.

## Figures and Tables

**Figure 1 micromachines-14-01436-f001:**
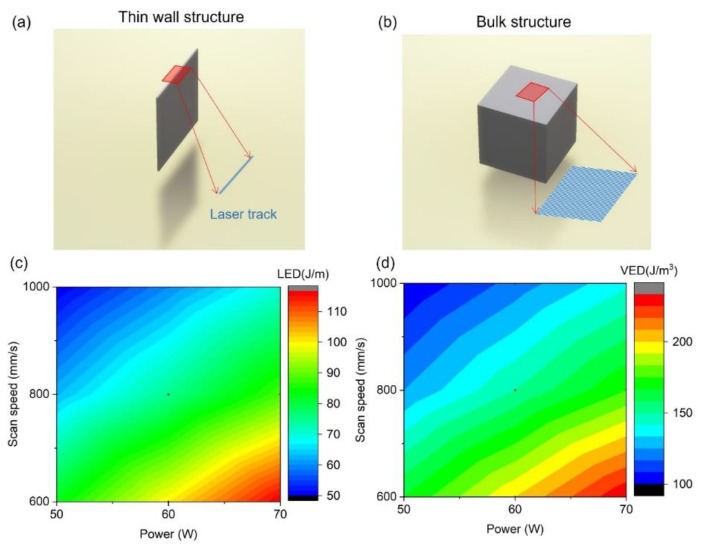
Designing models and scanning path of (**a**) thin wall structure, (**b**) bulk structure, and (**c**,**d**) LED and VED distribution with process parameters.

**Figure 2 micromachines-14-01436-f002:**
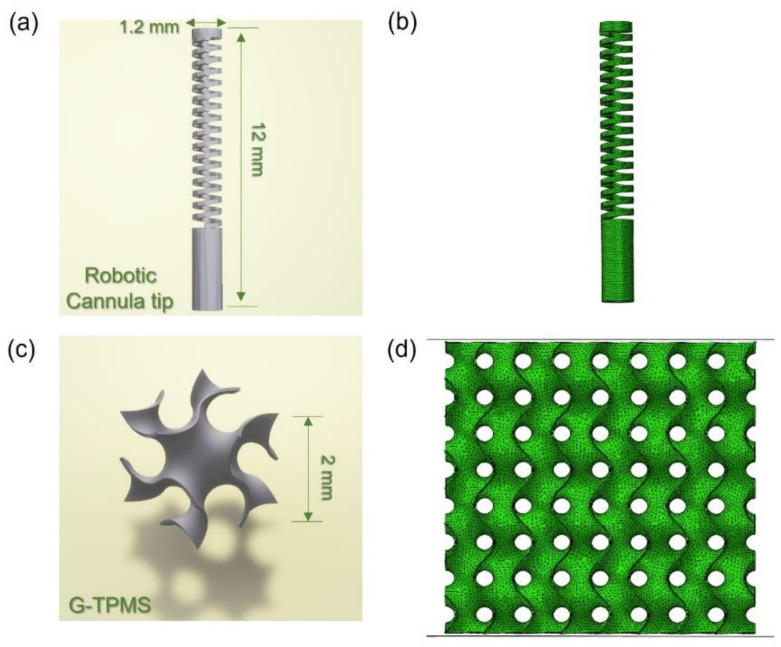
Models of NiTi components: (**a**) robotic cannula tip, (**b**) assembly of tip FE model, (**c**) G-TPMS lattice (unit cell), and (**d**) assembly of G-TPMS FE model.

**Figure 3 micromachines-14-01436-f003:**
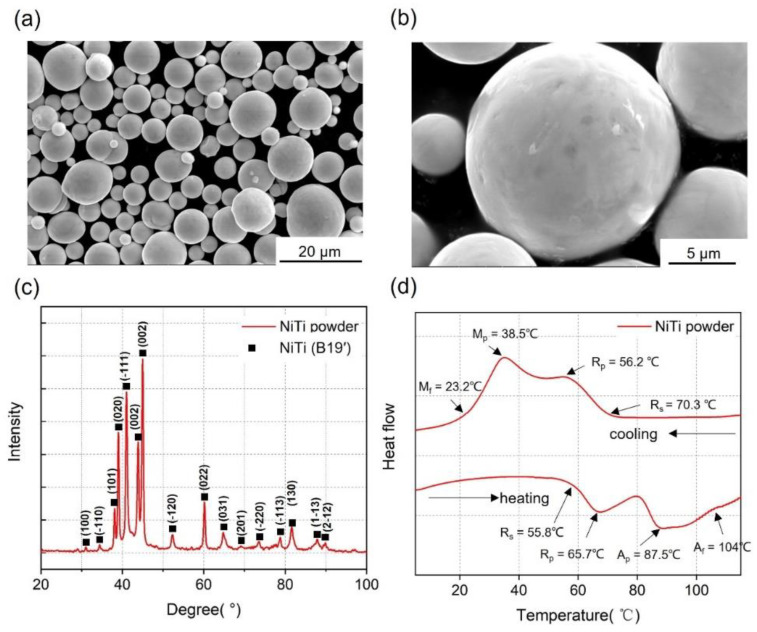
Characterization of NiTi original powder. (**a**,**b**) SEM observations at different magnifications, (**c**) XRD patterns, and (**d**) DSC curves.

**Figure 4 micromachines-14-01436-f004:**
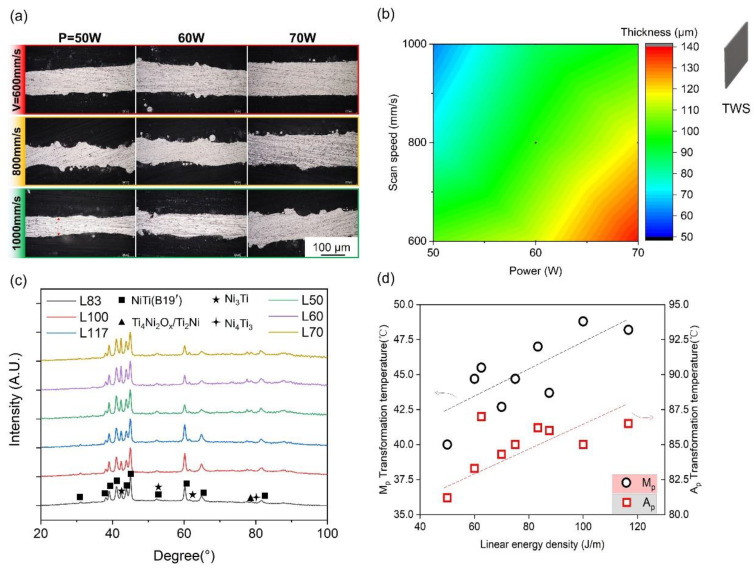
TWS characterizations: (**a**) Cross-sectional morphologies, (**b**) wall thicknesses of TWS samples, (**c**) XRD patterns, and (**d**) DSC analyses of phase transformation behaviors.

**Figure 5 micromachines-14-01436-f005:**
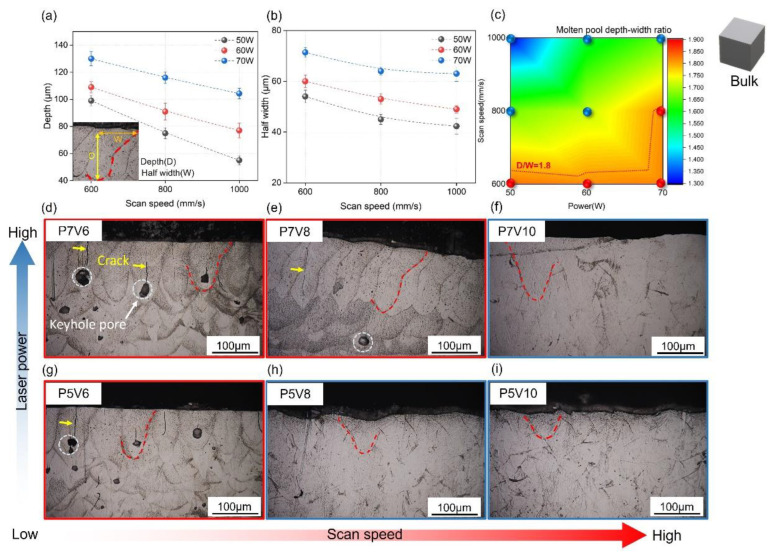
Microstructural characterization of cubes fabricated with different parameters. (**a**) Depth of molten pools. (**b**) Half-width of molten pools. (**c**) Depth-to-width ratios of molten pools. (**d**–**i**) Side surface morphologies of cubes.

**Figure 6 micromachines-14-01436-f006:**
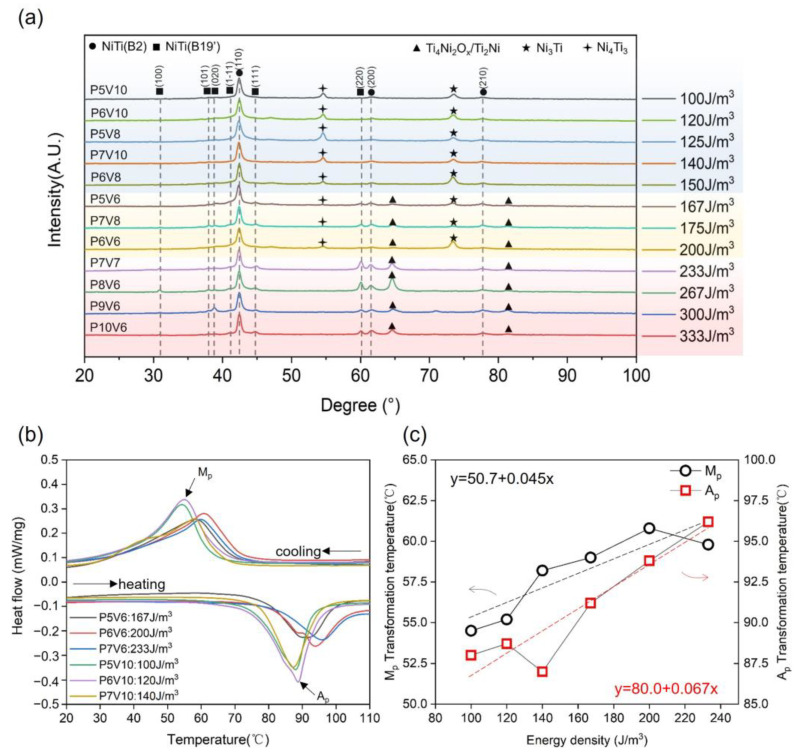
Phase compositions and transformation behaviors of as-printed samples using different parameters: (**a**) XRD profiles, (**b**) DSC curves, (**c**) Mp and Ap values determined from (**b**) as functions of VED.

**Figure 7 micromachines-14-01436-f007:**
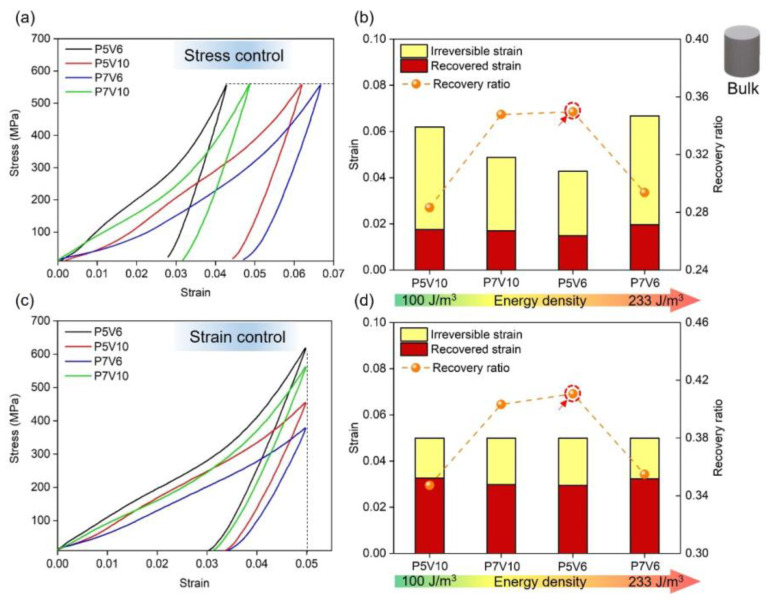
Shape recovery behavior of NiTi samples of P5V6/P5V10/P7V6/P7V10 (**a**) same stress compression, (**c**) same strain compression and (**b**,**d**) irreversible/ recovered strain and recovery ratio of corresponding samples.

**Figure 8 micromachines-14-01436-f008:**
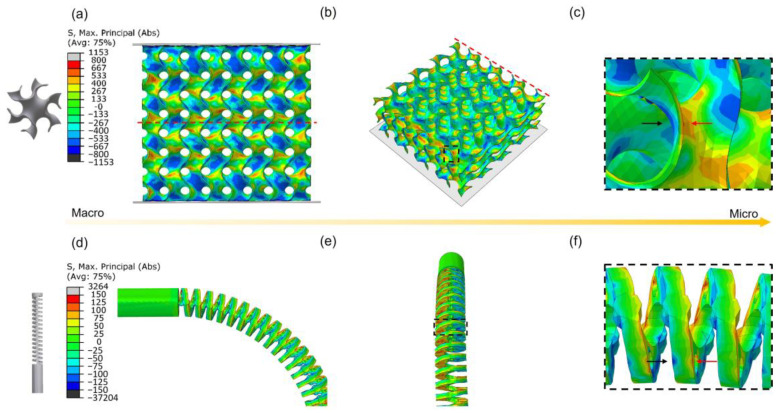
FE results of stress distribution of (**a**–**c**) G-type TPMS lattice under 6% compressed strain (**a**) front view, (**b**) cross-section view, (**c**) local surface, (**d**–**f**) cannula tip with a bend, (**d**) front view, (**b**) side view, and (**c**) local surface.

**Figure 9 micromachines-14-01436-f009:**
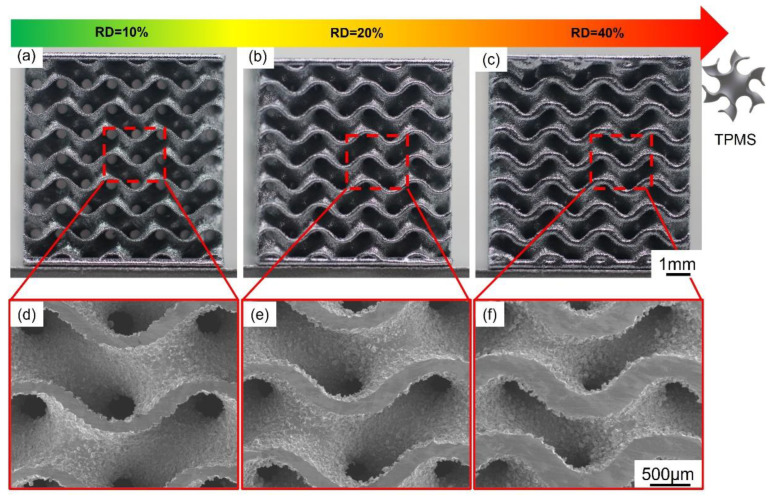
(**a**–**c**) Images of as-fabricated NiTi G-TPMS lattice structures with different relative densities of 10%/20%/40%, and (**d**–**f**) SEM images of corresponding TPMS lattices.

**Figure 10 micromachines-14-01436-f010:**
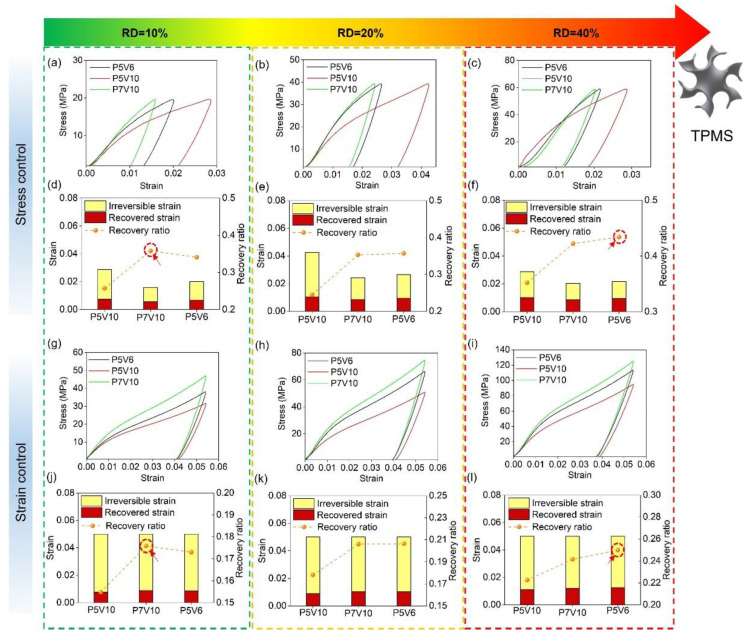
Shape recovery properties of G-TPMS structures, loading-unloading responses of different RD (10%, 20%, 40%) (**a**–**c**) at stress control, (**g**–**i**) at strain control, irreversible/recovered strain and recovered ratio of TPMS with different RD (**d**–**f**) at stress control, and (**j**–**l**) at strain control.

**Figure 11 micromachines-14-01436-f011:**
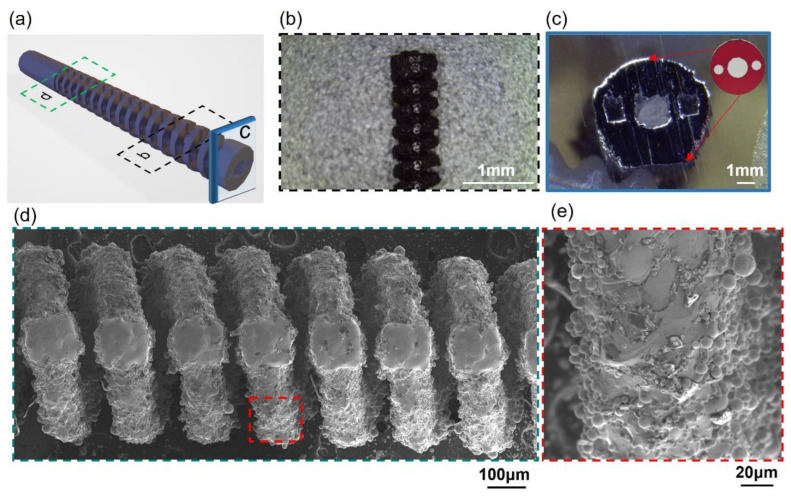
As-printed robotic cannula tip using P7V10 parameters: (**a**) Model, (**b**) top position by OM, (**c**) top cross-section by OM, (**d**) middle position by SEM, and (**e**) high-magnification SEM image of local position.

**Figure 12 micromachines-14-01436-f012:**
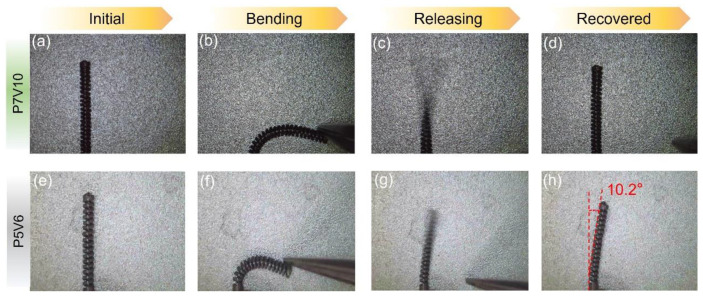
Superelastic properties of NiTi robotic cannula tip using P7V10 and P5V6 parameters at room temperature.

**Table 1 micromachines-14-01436-t001:** Particle size distribution of NiTi powder.

Property	Size	D10	D50	D90
Value	5–25 μm	10.0 μm	19.8 μm	28.5 μm

**Table 2 micromachines-14-01436-t002:** Chemical composition of NiTi powder.

Element	Ni	C	O	N	Fe	H	Ti
Value/wt%	55.2	0.01	0.07	0.002	0.01	0.001	Balance

**Table 3 micromachines-14-01436-t003:** Process parameters for thin-walled sample fabrication.

Sample Name	Laser Power (W)	Scan Speed (mm/s)	Linear Energy Density (J/m)
L50	50	1000	50
L60	60	1000	60
L63	50	800	63
L70	70	1000	70
L75	60	800	75
L83	50	600	83
L88	70	800	88
L100	60	600	100
L117	70	600	117

**Table 4 micromachines-14-01436-t004:** Process parameters for bulk sample fabrication.

Sample Name	Laser Power (W)	Scan Speed (mm/s)	Volumetric Energy Density (J/m^3^)
P5V6	50	1000	100
P6V6	60	1000	120
P7V6	50	800	125
P8V6	70	1000	140
P9V6	60	800	150
P10V6	50	600	167
P5V8	70	800	175
P6V8	60	600	200
P7V8	70	600	233
P5V10	80	600	267
P6V10	90	600	300
P7V10	100	600	333

**Table 5 micromachines-14-01436-t005:** Model parameters of TPMS samples.

Sample Name	Cell Size (mm)	Thickness (μm)	RD (%)
G10	2	65	10
G20	2	130	20
G40	2	260	40

## Data Availability

The data presented in this study are available on request from the corresponding author. The data are not publicly available due to project requirements.

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
