# Peer review of "Superelastic NiTi Functional Components by High-Precision Laser Powder Bed Fusion Process: The Critical Roles of Energy Density and Minimal Feature Size"

_micromachines, 2023, doi:10.3390/mi14071436_

Round 1

Reviewer 1 Report

The paper proposes a novel process optimization methodology and provides the processing guidelines for intricate NiTi components by L-PBF.

The authors performed a lot of work but, according to the reviewer’s point of view, the paper should be organized better; in particular, the "Materials and methods" section.

Some aspects should be improved in the manuscript before being considered for publication in MICROMACHINES. Accordingly, a review of the work should be performed.

-       Please, add the chemical composition of the NiTi alloy employed in this work. It is not enough to specify that it is a nearly equiatomic alloy.

-       Line 190: “configured” and not “conFigureured”.

-       The “Materials and methods” section presents also some results, such as XRD and DSC patterns. These are “results” of the initial characterization of the material so they should be moved to the “Results and discussion” section.

-       In equation (1), for most of the parameters the dimensional units are not specified.

-       The L-PBF process parameters should be added in the “Materials and methods” section, not as “supplementary material”. They are fundamental in the following discussion.

-       The authors want to add a lot of information, and for that use the strategy of “supplementary material”. The problem is that most of this material is fundamental for the comprehension and discussion of the work, so they should be added in the structure of the paper. Part of the supplementary material refers to the “Materials and method” part, even if they are mainly results. This is quite confusing and makes the paper less readable. Please improve the organization of the paper taking into account these suggestions.

The quality of English is fine.

Reviewer 2 Report

see attached

see attached.

Round 2

Reviewer 1 Report

The authors improved their paper by considering most of the reviewer's suggestions.

In the PDF file, there is a problem in the bibliography section with a double list of references that must be solved as a minor revision. 

Some minor editing of the English language is suggested.

Best regards.

The quality of the English language is quite good. 

Nevertheless, I suggest checking the English language as a minor revision.

Author Response

Many thanks for your constructive comments. The reference list has been revised accordingly. 

Reviewer 2 Report

accept.

Author Response

many thanks for your positive feedback on acceptance of our manuscript.